# The Relationship Between Changes in Physical Activity and Physical and Mental Health in Female Breast Cancer Survivors Undergoing Long-Term Activity Restrictions in Japan

**DOI:** 10.3390/nursrep15080279

**Published:** 2025-07-30

**Authors:** Naomi Tamai, Yasutaka Kimura, Ryuta Yoshizawa, Midori Kamizato

**Affiliations:** 1Department of Nursing, Meio University, Biimata, Nago 905-8585, Okinawa, Japan; y.kimura@meio-u.ac.jp (Y.K.); r.yoshizawa@meio-u.ac.jp (R.Y.); 2Department of Nursing, Okinawa Prefectural College of Nursing, Yogi, Naha 902-8513, Okinawa, Japan; kamizato@okinawa-nurs.ac.jp

**Keywords:** breast cancer survivor, COVID-19, physical activity, activity restrictions, nursing

## Abstract

**Purpose**: Exercise is recommended for survivors of breast cancer to alleviate adverse reactions and reduce the psychological burden. In recent years, however, environmental factors (e.g., pandemics and climate change) have made it difficult to exercise outdoors. Therefore, this study focused on the COVID-19 pandemic in Japan and evaluated the relationship between changes in physical activity (PA) and mental and physical health in breast cancer survivors. **Methods**: A questionnaire survey was conducted among 345 outpatient female breast cancer survivors aged between 29 and 69 years. The questionnaire was based on the International Physical Activity Questionnaire, the Patient Health Questionnaire-9, the European Organization for Research and Treatment of Cancer Core Quality of Life Questionnaire, and the Fear of COVID-19 Scale and included patient characteristics, changes in PA during pandemic restrictions, and needs for exercise support. The analysis categorized PA changes into two groups according to activity levels. The relationship between changes in PA and physical and mental health was evaluated using logistic regression analysis. **Results**: Patients with decreased PA accounted for 65.5% of the study population. Regardless of their activity level, these patients were aware of an increased susceptibility to COVID-19, showed a fear of the disease and a tendency for depression, and reported low life satisfaction and declined physical function. Of the patients who stopped exercising, 82.9% reported a decline in PA. Compared with those who had never exercised, those who stopped exercising saw their risk of depression increase by 15.6%. There was a high demand for personalized exercise support from healthcare professionals. **Conclusions**: Regardless of their activity level, decreasing PA during the pandemic decreased mental health and physical function in breast cancer survivors. There was a higher risk of depression among patients who stopped exercising. Because it is possible that similar situations may occur in the future, interventions by healthcare professionals must be considered in order to continue exercise.

## 1. Introduction

COVID-19 emerged in late 2019, with the causative SARS-CoV-2 virus spreading rapidly across the globe. In 2020, the World Health Organization [1] characterized the outbreak as a pandemic and declared a Public Health Emergency of International Concern. During the early stage of the pandemic, infection control measures, including a lockdown to restrict citizens’ mobility, were implemented all over the world. In Japan, measures including both voluntary and obligatory activity restrictions were put in place until May 2023, when the Japanese government reclassified COVID-19 as a “Class 5” disease according to the Infectious Disease Control Law. There were growing concerns that long-term activity restrictions would have adverse effects on people’s mental and physical health, including decreased physical activity (PA). Furthermore, in July 2023, UN Secretary-General António Guterres stated that “the era of global boiling has arrived”, sounding the alarm about climate change, including extreme heat [2]. In Japan, heatstroke alerts are issued in the summer, and outdoor exercise is restricted to prevent heatstroke. In this regard, the current situation is that it is difficult for breast cancer survivors to continue exercising outdoors due to infectious diseases and climate change. A similar situation is observed for other cancer survivors.

PA, including exercise, is considered beneficial for people with cancer. PA can improve physical health, including symptoms in cancer survivors, mental well-being, and quality of life (QOL) [3,4]. Therefore, there have been concerns about a decline in mental and physical functions [5,6] due to activity restrictions in patients living with cancer. Prior studies on the COVID-19 pandemic reported that decreased PA led to increased sitting time (90%), weight gain (5 ± 3.4 kg) [7], psychosocial impacts, and decreased QOL [8,9,10]. Furthermore, it was assumed that cancer survivors felt especially threatened by infection because of their weaker immune systems due to cancer treatment and had a higher risk of severe COVID-19 infection [11], resulting in a further decline in their PA levels.

As exercise is expected to improve prognosis and diminish adverse drug reactions in patients with breast cancer, the most common cancer among Japanese women, regular exercise is highly recommended [12]. In Japan, more than 97,800 new breast cancer cases are diagnosed each year, meaning that one in every nine Japanese women will be affected by the disease over their lifetime [13]. Breast cancer is a serious health issue in Japan. At present, the 5-year relative breast cancer survival rate is 99.3% for localized disease, but the rate decreases to 39.3% with distant metastases. The government has set a goal of 60% for cancer screening, but even in 2022, the current rate was 47.4%, one of the lowest among developed countries [14]. During the COVID-19 pandemic, the number of people who visited healthcare facilities decreased due to the closure of health check-up centers and the threat of infection [13]. The number of new breast cancer cases, which had been steadily increasing, decreased by 6.4% in 2020 compared to the previous year for the first time [15]. Thus, the pandemic had a significant impact on society, with many patients losing the opportunity for early cancer detection.

This study focused on the COVID-19 pandemic in Japan, aiming to evaluate the relationship between changes in physical activity (PA) and physical and mental health in breast cancer survivors undergoing long-term activity restrictions. There may be another pandemic in the future. By elucidating the relationship between PA and physical and mental health during the pandemic, it may be possible to provide breast cancer survivors with PA promotion support, allowing them to maintain their physical and mental wellness without being affected by the pandemic. The investigation was conducted in the Okinawa prefecture in Japan, where the spread of the COVID-19 infection was the most severe [16].

## 2. Materials and Methods

### 2.1. Study Design

This was a cross-sectional study. The survey was conducted during the largest pandemic waves in Japan, the 6th and 7th waves, between February and July 2022.

The Okinawa Prefecture, where our survey was conducted, is an island prefecture located in the southwest of Japan with a population of 1.457 million. It had the highest number of COVID-19 cases per 100,000 population in Japan and is a region where the spread of infection was serious, and medical support was requested from other regions and the military (as of 14 August 2021, the number of new cases was 752 per day, the number of infected cases per 100,000 population was 283.64, and the number of effective infected cases was 1.14) [16]. The survey was conducted at the outpatient units of five hospitals in the prefecture, which had the highest numbers of breast cancer cases.

The survey sample size was set at approximately 400 patients based on previous data indicating that 41.6% of breast cancer survivors do not exercise [17]; the number of individuals required to calculate the 95% confidence interval at 15% was 161; the number of individuals required to perform a regression analysis of 14 items in the multivariate analysis was 140 [18]; and the response rate for patients with breast cancer at the outpatient unit was estimated at 40%.

The subjects were selected by having the target facilities recommend people with no restrictions on exercise, and the researchers met with them in person to request the subjects. They were free to choose to collect the data via collection boxes, mail, or the web. The researchers waited in the outpatient clinic to be able to answer any questions and distributed request forms with their contact information.

### 2.2. Participants and Recruitment Strategy

To assess PA in breast cancer survivors during the COVID-19 pandemic, we analyzed the questionnaire responses of 345 female patients aged between 20 and 69 years based on the physical activity levels defined by the Japanese Ministry of Health, Labour and Welfare [19] to eliminate sex and age differences (Figure 1). The selection criteria were as follows: females aged between 20 and 69 years with a breast cancer diagnosis who had received no exercise restrictions from their primary doctors and who were able to respond at their own will.

This study employed an anonymous self-administered questionnaire survey. The candidate participants were selected by nurses if they met the selection criteria. The researchers invited the candidates to participate in the survey in person and/or in writing, with a maximum response completion time of two weeks. The participants could also choose between an online or mail-back survey according to their preferences. Consent was deemed to have been given by submitting a response form, answering online, or sending a response by post.

### 2.3. Demographic and Clinical Characteristics

The demographic data (age, marital status, family members in the household, educational background, and occupational history) and the clinical characteristics (time since diagnosis, disease staging, treatment status, and symptoms) of the participants were analyzed.

### 2.4. Physical Status

We investigated the exercise habits of participants before the COVID-19 pandemic, and for those who stopped exercising during the pandemic, the reasons for stopping were examined.

Changes in PA before the pandemic and at the time of the survey were evaluated using a 5-point Likert scale from 1 (became less active) to 5 (became more active), which is a subjective measure. We categorized PA changes into two groups: the “decreased PA” group for participants who answered 1 or 2, and the “maintained PA” group for those who answered 3 to 5.

The activity levels were calculated based on the International Physical Activity Questionnaire (IPAQ), an objective indicator [20], with the number of days and time spent performing PA and the intensity of PA being calculated as metabolic equivalent task (MET) values. MET-min/week was classified into three categories: low (<700 MET), moderate (700–2519 MET), and high (>2520 MET). Activity levels were classified into two groups, with those with low PA levels classified as the “low-IPAQ group” and those with moderate and high PA levels as the “moderate–high-IPAQ group”.

Changes in body weight before the pandemic and at the time of the survey were evaluated on a three-point scale: increased, no change, and decreased.

### 2.5. Infection Prevention Measures During COVID-19 and the Fear of COVID-19 Scale (FCV-19S)

COVID-19 infection prevention measures, awareness of susceptibility to COVID-19 infection, and the reasons for them were examined. In addition, we used the Japanese version [21] of the FCV-19S developed by Ahorsu et al. [22] to quantitatively assess COVID-19 fear. FCV-19S consists of seven items, each rated on a 5-point scale, with 1 corresponding to “strongly disagree” and 5 to “strongly agree” and a maximum total score of 35 points. The higher the score, the greater the level of COVID-19 fear.

### 2.6. Psychological Status

To evaluate depression, we used the Japanese version of the Patient Health Questionnaire-9 (PHQ-9) [23]. Developed by Kroenke et al. [24] as a screening test of depression, the patient questionnaire consists of nine items, each rated on a 4-point scale, with a maximum total score of 36. Severity categories are as follows: no depression, 1 to 4; mild depression, 5 to 9; moderate depression, 10 to 14; moderately severe depression, 15 to 19; and severe depression, 20 or more. The optimal cut-off score for diagnosing depression is 10 or more.

The Japanese version of the European Organization for Research and Treatment of Cancer (EORTC QLQ-C30) [25], originally developed by Aaronson et al. [26], was used to evaluate quality of life, function, and symptoms in patients with cancer. The questionnaire incorporates general QOL scales, functional scales (physical, role, cognitive, emotional, and social functions), and symptom scales (fatigue, nausea, pain, dyspnea, loss of appetite, insomnia, constipation, diarrhea, and financial difficulties). All scales and single-item measures range in score from 0 to 100. A high scale score represents a higher response level. Thus, a high score for a functional scale/item represents a high/healthy level of functioning, while a high score for a symptom scale/item represents a high level of symptomatology/problems. The score of each item was calculated based on EORTC version 3.0 [27].

Life satisfaction during the COVID-19 pandemic was assessed using a 10-point scale developed by Tang et al. [28] ranging from “0: very dissatisfied” to “10: very satisfied”. The change in life satisfaction observed at the time of the survey in comparison to before the pandemic was evaluated on a 5-point scale ranging from “1: much worse” to “5: much better”.

The relationship between changes in PA and physical and mental health was assessed for each group.

### 2.7. Demand for PA Support Before and During the COVID-19 Pandemic

The demand for exercise support by breast cancer survivors was evaluated on a 7-point scale ranging from “1: not necessary at all” to “7: very necessary”. The relationship with PA change was analyzed on a scale of “not necessary (1–4 points)” and “very necessary (5 points or more)”. To compare the data before and during the pandemic, we compared it with data examined by the authors in 2018.

### 2.8. Statistical Analysis

Categorical variables were summarized using frequencies and percentages, and means and standard deviations were calculated for continuous variables. The chi-square test was used to compare qualitative data, the *t*-test was used to compare means of normally distributed quantitative data, and non-parametric tests were used for non-normal quantitative data. The Pearson correlation coefficient was calculated to assess the strength and direction of the linear relationship between normally distributed variables. For non-ordinal variables, Spearman’s correlation coefficients were calculated. Items that showed significant differences in the correlation analysis were used as independent variables in the logistic regression analysis.

To examine factors related to PA changes during the COVID-19 pandemic, the changes in PA were considered the dependent variables, and the 14 items that showed significant differences in univariate analysis were considered the independent variables. Stepwise logistic regression analysis was performed, and the relationship between the changes in PA and physical and mental health was extracted.

All statistical analyses were conducted using IBM SPSS Statistics 25.0, and a *p*-value < 0.05 was considered statistically significant.

### 2.9. Ethics

Ethical considerations included obtaining informed consent, respecting free will, protecting privacy, and complying with infection control measures at the target facilities. The study was conducted with the approval of the ethical review committees of the affiliated and target facilities. Approval was granted by the Research Ethics Review Committee of Meio University (Approval No. 2021-053, date; 23 December 2021) and ethics committees of the research facilities.

## 3. Results

### 3.1. Participants

The characteristics of the participants are listed in Table 1. The mean age was 55.7 ± 8.25 years, and the time since diagnosis was 71.4 ± 63.4 months. Of the patients, 40.6% (140) were in stage 0 to 1, 237 (68.7%) were undergoing treatment, and 236 (68.4%) were symptomatic.

### 3.2. Physical Status and Physical Activity

There were 172 patients with exercise habits (49.9%) before the pandemic, of whom 78 (45.6%) stopped exercising. Forty-seven patients (60.3%) stopped exercising for reasons related to infection prevention measures, such as “no longer exercising with the view to prevent infection” and “lost access to spaces for exercising due to infection prevention measures, such as gym closures”. PA decreased in 226 patients (65.5%) and was maintained in 119 patients (34.5%) (Table 2).

Regarding IPAQ activity levels, 162 patients (47.0%) were categorized in the low-IPAQ group and 183 patients (53.0%) in the moderate–high-IPAQ group (Table 3). Approximately 60% of the patients experienced decreased PA in both groups (low-IPAQ group: 109 patients [67.3%]; moderate–high-IPAQ group: 117 patients [63.9%]).

### 3.3. COVID-19 Infection Prevention Measures and Psychological Aspects According to PA Level

Over 80% of patients took COVID-19 infection prevention measures (Table 3). Those with decreased PA had higher awareness of susceptibility to infection in both IPAQ groups (low-IPAQ group: *p* < 0.001; moderate–high-IPAQ group: *p* = 0.006) with the following reasons: “cancer (*p* < 0.001)”, “impact of cancer treatment (*p* < 0.001)”, “impact of treatment other than cancer (*p* = 0.002)”, and “age (*p* = 0.002)”.

The mean value of FCV-19S, a COVID-19 fear indicator, was 16.9 ± 5.2 (Table 4). The FCV-19S score was higher in participants with decreased PA regardless of their activity level (low-IPAQ group: *p* = 0.023; moderate–high-IPAQ group: *p* < 0.001), indicating that they felt threatened by COVID-19.

The score of the global health status/QOL scale, an EORTC subscale, was lower in participants with decreased PA in all activity levels, and it was significantly low in the low-IPAQ group (*p* < 0.001). All the items in the functional scales had a low score in participants with decreased PA regardless of their activity level, showing a significant difference in physical function (low-IPAQ group: *p* = 0.032; moderate–high-IPAQ group: *p* = 0.011). Participants with decreased PA, regardless of their activity level, scored a higher level of symptomatology in all the symptom items, especially fatigue (low-IPAQ group: *p* = 0.004; moderate–high-IPAQ group: *p* = 0.021).

Life satisfaction at the time of the survey was lower in participants with decreased PA regardless of their activity level (low-IPAQ group: *p* = 0.003; moderate–high-IPAQ group: *p* = 0.001). Moreover, there were more people whose life satisfaction worsened during the COVID-19 pandemic amongst participants with decreased PA (*p* < 0.001).

### 3.4. PA Influencing Factors During COVID-19

There were two factors related to the changes in PA, namely the depression module in PHQ-9 (OR = 0.844; 95% CI 0.759–0.938; *p* = 0.002) and stopping exercising due to COVID-19 (OR = 0.171; 95% CI 0.038–0.771; *p* = 0.022), with an accuracy rate of 69.0% (Table 5). PA decreased by 82.9% in participants who stopped exercising due to COVID-19, and the risk of depression increased by 15.6% in those with decreased PA compared with those with sustained PA.

### 3.5. Results of Demand for PA Support Before and During the COVID-19 Pandemic

The PA support items in highest demand were “exercise at home” with 298 entries (86.4%), followed by “information provided by brochures” with 244 entries (70.7%) and “information provided by online materials” with 237 entries (68.7%), indicating a high demand for support overall regardless of changes in PA (Table 6). To compare the data before and during the pandemic, we compared it with data examined by the authors in 2018. Individual support by healthcare professionals was a demand of about 60% of participants, especially those who maintained PA (*p* = 0.009–0.036).

## 4. Discussion

To the best of our knowledge, this is the first study in Japan to reveal a relationship between changes in PA and physical and mental health in breast cancer survivors during the COVID-19 pandemic and risk factors based on a multifaceted assessment including psychosocial factors.

The proportion of breast cancer survivors whose PA decreased during the COVID-19 pandemic was 65.5%, whereas the proportion of those with increased PA was merely 3.5%. Almost half (45.6%) of the patients who had no exercise habits at the time of the survey had stopped exercising during the pandemic, and 60.3% of them did so for reasons related to COVID-19. In the EORTC function scales, participants with decreased PA had lower scores in all items. There was a significant difference in physical function between the two IPAQ groups. Consistent with previous reports [7,8,9,10,29,30], this study also shows that a decrease in PA may lead to a decrease in physical function.

Participants who were aware of their susceptibility to COVID-19 were more likely to have low PA regardless of their activity level, and the reason for this included cancer and cancer treatment. In a previous study [17], 119 patients (41.3%) reported their activities to have changed since their breast cancer diagnosis and reported “becoming less active”. In this study, the awareness of susceptibility to COVID-19 and the threat of infection, along with cancer and cancer treatment, likely contributed to the decline in PA observed in breast cancer survivors.

Furthermore, 68.4% of the patients were symptomatic, with fatigue being the most common symptom (40.0%). In the EORTC symptom scales, those with decreased physical function showed a significantly higher symptomatic level of fatigue regardless of their activity level. Tabaczynski et al. [10] reported that PA decreased during the pandemic, which may have negative implications for cancer survivors in terms of QOL and fatigue. A relationship between PA and fatigue has previously been reported [31], but the results of this study show that regardless of their activity level, decreased PA can lead more patients to become symptomatic, with a higher level of symptomatology including fatigue. To alleviate the symptoms, it is important to maintain the current activity level, whatever it may be.

Regardless of their activity level, participants with decreased PA also had higher scores for the PHQ, a questionnaire module for depression, indicating a tendency towards moderate to severe depression. Seven et al. [32] showed that breast cancer survivors experienced different health challenges causing new physical and psychological symptoms, such as lymphedema, pain, burnout, and anxiety, which may have long-term effects on their QOL. This study also showed that breast cancer survivors with decreased PA were aware of their susceptibility to COVID-19 and felt threatened due to their cancer and its treatment, resulting in reduced QOL. Furthermore, many of those with decreased PA, regardless of their activity level, reported that their life satisfaction had worsened, indicating that decreased PA is associated with an increased risk of depression and decreased QOL and life satisfaction.

Consequently, it was revealed that decreased PA in breast cancer survivors, regardless of their activity levels, has an impact on their physical and mental health. Tabaczynski et al. [10] concluded that behavioral strategies are needed to help cancer survivors to engage in and maintain PA.

In terms of support needs, 86.4% of participants requested “exercise that can be done at home”, followed by 70.7% of participants that desired “information provided through brochures”. In comparison with the period preceding the COVID-19 pandemic [17], more breast cancer survivors requested remote and home support rather than face-to-face support. In addition, those with decreased PA tended to request remote support by phone calls and emails, whereas those with maintained PA sought personalized support from healthcare professionals. Previously, Sagarra-Romero et al. [33] reported an online home-based exercise intervention during COVID-19 lockdowns that could improve physical fitness and body composition in breast cancer survivors. Moreover, Blasio et al. [34] demonstrated that personalized feedback and supervised online physical exercise sessions conferred major physical activity benefits in breast cancer survivors during the COVID-19 pandemic. Therefore, remote support should be considered during pandemics. However, exercise intervention usually has a short-term effect, and in the longer term, many survivors may become insufficiently active [35]. Breast cancer survivors may need continued motivation and practical support tailored to their individual characteristics and physical activity history to incorporate exercise into an everyday routine in the long term [36].

The main results of this study were the PA changes observed in breast cancer survivors during the COVID-19 pandemic and the most recent distributions of QOL and depression rates among this patient population. Interestingly, a decrease in PA, rather than the activity level per se, affected the mental and physical health of patients. Of particular importance, the risk of depression increased when patients who used to exercise stopped exercising.

Various infectious diseases, including COVID-19, have persisted even today. Hence, it is essential to develop personalized exercise support provided by medical professionals that is unaffected by pandemics so that breast cancer survivors can continue exercises, in particular a healthcare team consisting of physicians and rehabilitation therapists that share information about the patient’s medical condition and effective exercise methods and provide individualized exercise support that is rooted in the patient’s daily life.

Because it is possible that similar situations may occur in the future and considering climate change, the findings of this study have important implications for the care of breast cancer survivors.

### Study Limitations

This survey was a cross-sectional study that was performed between February and July 2022, two years after the pandemic started. Additionally, cross-sectional studies cannot prove causality. The pandemic period itself may have impacted breast cancer survivors’ mental health, including loss of independence, loss of family members, and confinement. In addition, because this study focused on changes in physical activity, only women were analyzed to avoid gender bias, limiting generalizability. Although it has been reported that sexual dysfunction affects mental problems and QOL in female breast cancer survivors [37], this study did not examine the relationship with sexual dysfunction. Therefore, the applicability of the results is limited. However, the study results can help establish exercise support that is unaffected by the environment.

## 5. Conclusions

Breast cancer survivors with decreased PA were aware of their susceptibility to COVID-19 and felt under threat, resulting in decreased life satisfaction during the pandemic. Participants with decreased PA were more likely to show a tendency towards depression, and the risk of depression increased when people who used to exercise stopped exercising. Based on these results, promoting PA during the COVID-19 pandemic could have reduced the risk of depression and improved the QOL of breast cancer survivors. This issue is especially serious in patients who have exercise habits and lose them due to a lack of space for exercising. Therefore, intervention by healthcare professionals must be urgently considered to promote exercise in a pandemic situation.

## Figures and Tables

**Figure 1 nursrep-15-00279-f001:**
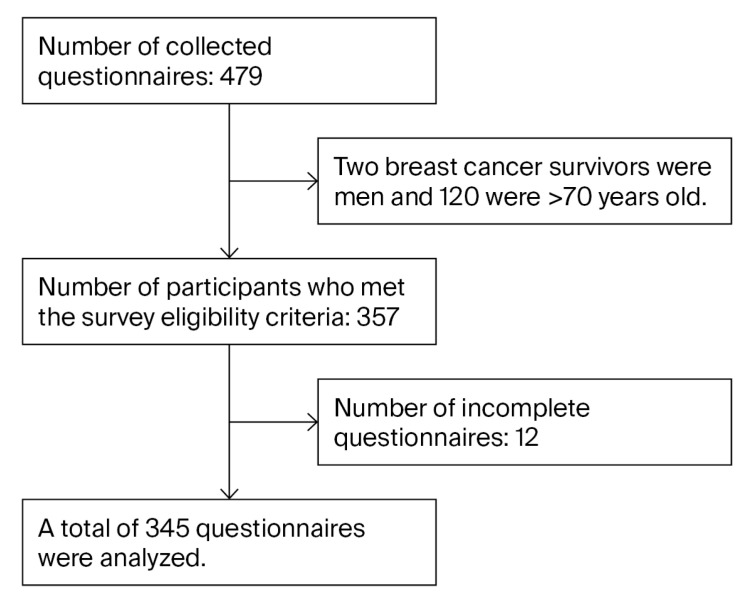
Flow diagram of study participants.

**Table 1 nursrep-15-00279-t001:** Characteristics of breast cancer survivors (*n* = 345).

Variable	Mean ± SD	*n* (%)
Demographic		
Age (years)	55.7	±8.3
Marital status		
Single	49	(14.2)
Married	249	(72.2)
Widowed/divorced	45	(13.0)
Undisclosed	2	(0.6)
Family members in the household		
Living alone	48	(13.9)
Two	125	(36.2)
Three or more	172	(49.9)
Educational attainment		
Partial/full high school completion	132	(38.3)
Partial/full specialized high school completion	42	(12.2)
Partial/full university/college completion	161	(46.7)
Undisclosed	4	(1.2)
Employment		
Full-time	145	(42.0)
Part-time	88	(25.5)
Retired/no profession	106	(30.7)
Undisclosed	6	(1.7)
Medical		
Time since diagnosis (months)	71.4	±63.4
Stage at diagnosis		
Stages 0–I	140	(40.6)
Stage II	96	(27.8)
Stage III	23	(6.7)
Stage IV	11	(3.2)
Unsure	42	(12.2)
Treatment status		
Completed treatment	91	(26.4)
Undergoing treatment	237	(68.7)
Chemotherapy	24	(7.0)
Radiotherapy	8	(2.3)
Hormone therapy	224	(64.9)
Symptoms		
No	101	(29.3)
Yes	236	(68.4)
Fatigue	138	(40.0)
Stiff joints	92	(26.7)
Edema	63	(18.3)
Numbness	53	(15.4)
Anxiety	52	(15.1)
Insomnia	52	(15.1)
Pain	38	(11.0)
Alopecia	27	(7.8)
Other symptoms	17	(4.9)

**Table 2 nursrep-15-00279-t002:** Physical Status and Physical activity.

Variable	Mean ± SD	*n* (%)
Exercise habits: Yes	172	(49.9)
No (includes 78 people who stopped exercising)	171	(49.6)
Reason for stopping exercise (*n* = 78)	78	(45.6)
No time for exercise	33	(42.3)
Exercise stopped as an infection control measure	26	(33.3)
No place to exercise due to infection control measures	11	(14.1)
Physical activity during the COVID-19 pandemic		
PA↓: decreased physical activity	226	(65.5)
Less active	70	(20.3)
Somewhat inactive	156	(45.2)
PA→: physical activity was maintained or improved	119	(34.5)
No change	107	(31.0)
More active	12	(3.5)
Weight change before and during the COVID-19 pandemic		
Weight gain (kg)	3.27	±2.2
Weight loss (kg)	3.76	±2.0

PA, physical activity.

**Table 3 nursrep-15-00279-t003:** Results of the International Physical Activity Questionnaire (IPAQ).

	*n* = 345		Low-IPAQ Group (*n* = 162)		Moderate–High-IPAQ Group (*n* = 183)		*p*-Value
	Mean ± SD	*n* (%)	Mean ± SD	*n* (%)	Mean ± SD	*n* (%)	
Low-IPAQ group	162	(47.0)	-		-		
Moderate-IPAQ group	125	(36.2)	-		-		
High-IPAQ group	58	(16.8)	-		-		
Physical change before and during the COVID-19 pandemic							
PA was decreased (PA↓)	226	(65.5)	109	(67.3)	117	(63.9)	0.514
PA was maintained or increased (PA→)	119	(34.5)	53	(32.7)	66	(36.1)	
High-level PA:							
Vigorous-intensity PA per week (yes)	70	(20.3)	8	(5.2)	62	(34.6)	<0.001
Total hours (minutes/day) ^§^	74.1	±73.5	45.7	±42.3	77.4	±75.8	0.118
Vigorous METs^§^	337.6	±1258.5	7.4	±54.3	629.9	±1675.8	<0.001
Moderate-level PA:							
Moderate-intensity PA per week (yes)	113	(32.8)	17	(11.0)	96	(54.5)	<0.001
Total hours (minutes/day) ^§^	111.9	±243.1	45.1	±48.0	125.1	±263.4	0.007
Moderate METs ^§^	515.7	±2456.6	19.7	±79.6	954.7	±3314.9	<0.001
Walk:							
Walk every week (yes)	265	(76.8)	95	(60.1)	170	(93.9)	<0.001
Walking time (min/day) ^§^	90.5	±129.2	33.8	±19.9	118.0	±149.4	<0.001
Walking METs ^§^	1107.9	±2443.8	154.5	±1953	1952.0	±3119.1	<0.001
Total PA (METs—min/wk) ^§^	1961.2	±4069.2	181.6	±204.5	3536.6	±5093.7	<0.001
Sedentary time (min/wk) ^§^	354.2	±255.5	393.0	±278.2	320.6	±229.5	0.009

Chi-square (χ^2^) test and *t*-test ^§^. Analysis was performed by excluding “Unknown” answers. Moderate–high-IPAQ group includes both vigorous- and moderate-PA groups. PA, physical activity; IPAQ, International Physical Activity Questionnaire.

**Table 4 nursrep-15-00279-t004:** Outcomes of physical activity for breast cancer survivors during the COVID-19 pandemic.

	Number of Participants (*n* = 345)		Low-IPAQ Group (*n* = 162)				*p*-Value	Moderate–High-IPAQ Group (*n* = 183)				***p*-Value**
			PA↓ (*n* = 109)		PA→ (*n* = 53)			PA↓ (*n* = 117)		PA→ (*n* = 66)		
	Mean ± SD	*n* (%)	Mean ± SD	*n* (%)	Mean ± SD	*n* (%)		Mean ± SD	*n* (%)	Mean ± SD	*n* (%)	
Infection control measures (multiple answers)												
Wearing a mask	340	(98.6)	106	(97.2)	51	(96.2)	0.724	117	(100.0)	66	(100.0)	-
Hand hygiene	316	(91.6)	103	(94.5)	46	(86.8)	0.090	109	(93.2)	58	(87.9)	0.224
Vaccination	296	(85.8)	95	(87.2)	46	(86.8)	0.948	105	(89.7)	50	(75.8)	0.012
Avoiding crowds	288	(83.5)	91	(83.5)	41	(78.8)	0.474	132	(88.0)	53	(80.3)	0.157
Refraining from going out	184	(53.3)	66	(60.6)	24	(45.3)	0.067	73	(62.4)	21	(31.8)	<0.001
Recognized vulnerabilities to COVID-19												
Yes	62	(18.0)	29	(27.4)	8	(15.4)	<0.001	22	(19.3)	3	(4.5)	0.006
No	276	(80.0)	77	(72.6)	44	(84.6)		92	(80.7)	63	(95.5)	
No response	7	(2.0)										
FCV-19S (1–35) ^§^	16.9	±5.2	17.8	±5.6	16.0	±4.9	0.038	17.5	±5.1	15.1	±4.7	0.001
COVID-19 scares me more than anything else. (1–5) ^§^	3.3	±1.0	3.4	±1.0	3.2	±1.1	0.345	3.4	±1.0	3.1	±1.0	0.007
I’m not comfortable thinking about COVID-19. (1–5) ^§^	2.4	±0.9	2.5	±1.0	2.2	±0.8	0.043	2.5	±0.9	2.1	±0.9	0.002
My hands are sweating when I think of COVID-19. (1–5) ^§^	1.7	±0.8	1.8	±1.0	1.7	±0.8	0.297	1.6	±0.7	1.5	±0.8	0.093
I’m afraid I’m going to die from COVID-19. (1–5) ^§^	3.4	±1.3	3.4	±1.3	3.3	±1.3	0.535	3.4	±1.1	3.1	±1.4	0.073
I get nervous or anxious when I see news or talk about COVID-19 on social media. (1–5) ^§^	2.6	±1.1	2.7	±1.2	2.4	±1.2	0.191	2.8	±1.0	2.3	±1.1	0.003
I can’t sleep because I’m worried about COVID-19 infection. (1–5) ^§^	1.7	±0.8	1.8	±0.8	1.5	±0.6	0.020	1.7	±0.8	1.4	±0.7	0.011
The thought of COVID-19 infection makes my heart skip a beat. (1–5) ^§^	2.1	±1.1	2.2	±1.1	1.9	±1.0	0.047	2.2	±1.1	1.8	±1.0	0.013
PHQ-9												
Mean PHQ-9 score (0–27) ^§^	4.0	±4.3	5.0	±5.0	2.4	±2.8	<0.001	4.3	±4.0	3.1	±4.1	0.062
Depression symptoms (%)												
None (0 to 4)	226	(65.5)	60	(56.1)	42	(82.0)	0.004	71	(62.3)	53	(80.3)	0.042
Mild (5 to 9)	79	(22.9)	33	(30.8)	8	(16.0)		29	(25.4)	9	(13.6)	
Moderate (10 to 27)	33	(9.5)	14	(13.1)	1	(2.0)		14	(12.3)	4	(6.0)	
EORTC												
Global health/QOL (0–100) ^§^	64.9	±21.6	59.0	±21.4	72.8	±18.1	<0.001	64.2	±22.2	69.6	±20.9	0.107
Functional scales (0–100)												
Physical functioning ^§^	90.8	±11.2	88.6	±12.3	92.7	±9.9	0.032	90.0	±12.1	94.3	±7.4	0.011
Role functioning ^§^	90.3	±17.0	85.5	±20.1	94.1	±12.4	0.006	90.5	±17.0	94.9	±12.5	0.072
Emotional functioning ^§^	86.6	±16.8	83.6	±18.5	89.5	±13.3	0.024	85.7	±16.8	90.5	±15.7	0.062
Cognitive functioning ^§^	84.1	±17.9	82.7	±18.2	85.9	±15.8	0.254	82.5	±18.9	88.2	±16.6	0.036
Social functioning ^§^	91.0	±17.1	88.0	±19.2	91.8	±19.3	0.239	91.2	±15.6	95.4	±13.0	0.066
Symptom scales (0–100)												
Fatigue ^§^	27.5	±22.6	33.8	±23.7	23.3	±19.7	0.004	27.8	±23.2	20.3	±19.2	0.021
Nausea and vomiting ^§^	1.4	±5.7	1.9	±5.8	0.7	±3.3	0.163	1.5	±6.1	1.3	±6.1	0.836
Pain ^§^	12.8	±19.8	18.1	±22.2	7.7	±17.6	0.004	12.0	±20.0	9.5	±14.4	0.378
Dyspnea ^§^	8.2	±16.0	11.1	±18.2	5.9	±12.8	0.067	7.8	±16.1	6.2	±14.3	0.473
Insomnia ^§^	21.3	±28.0	25.3	±28.8	14.0	±23.4	0.010	22.6	±29.5	18.5	±26.4	0.333
Appetite loss ^§^	7.0	±15.1	8.9	±17.4	3.9	±12.7	0.068	8.4	±15.8	3.6	±10.3	0.027
Constipation ^§^	17.7	±32.7	24.4	±39.9	13.1	±20.1	0.019	16.2	±35.4	12.3	±19.2	0.336
Diarrhea^§^	7.1	±15.2	9.6	±19.4	6.5	±13.4	0.314	5.2	±12.2	7.2	±13.8	0.341
Financial difficulties ^§^	12.4	±24.0	14.2	±24.2	11.8	±27.3	0.589	11.6	±22.5	10.8	±24.4	0.823
Life satisfaction (1–10) ^§^	7.0	±1.9	6.5	±2.1	7.5	±1.6	0.003	6.8	±1.9	7.7	±1.7	0.001
Satisfaction with life changes before and during the COVID-19 pandemic												
Very poor	162	(47.0)	70	(64.2)	14	(26.4)	<0.001	66	(57.4)	12	(18.2)	<0.001
Neither poor nor good	159	(46.1)	34	(31.2)	33	(62.3)		42	(36.5)	50	(75.8)	
Good	20	(5.8)	3	(2.8)	6	(11.3)		7	(6.1)	4	(6.1)	
Non-response	4	(1.6)										

PA↓ represents the group with decreased PA. PA→ represents the group with maintained or improved PA. The change in quality of life before and after the COVID-19 pandemic was rated on a 10-point scale from 1 (“very dissatisfied”) to 10 (“very satisfied”). The *t*-test ^§^ and χ^2^ test were used. The analysis excluded non-responses. The overall mean of the PHQ, a depression module, was 4.0 ± 4.3 (Table 3). The score was significantly higher at 5.0 ± 5.0 (equivalent to moderate depression) in participants with decreased PA in the low-IPAQ group (*p* < 0.001). Moreover, moderate and severe depression was significantly more frequent in participants with decreased PA regardless of their activity level (low-IPAQ group: *p* < 0.001; moderate–high-IPAQ group: *p* = 0.042).

**Table 5 nursrep-15-00279-t005:** Factors associated with changes in physical activity during the COVID-19 pandemic.

	Partial Regression Coefficient	Standard Error	*p*-Value	OR	(95% CI)
PHQ-9	−0.171	0.055	0.002	0.844	(0.759–0.938)
Exercise stopped as an infection control measure	−1.768	0.769	0.022	0.171	(0.038–0.771)

Binomial logistic regression analysis (forward stepwise). The dependent variable was the change in physical activity (PA) (1: PA↓, 2: PA→). The independent variables were the items with significant differences in the univariate analysis. PHQ-9 score (0–27; the higher the score, the more severe the depressive symptoms). Stopped exercising to prevent infection (0, no; 1, yes). Significant probability of Hosmer–Lemeshow χ^2^ statistic: 2.434; *p* = 0.932; accuracy rate: 69.0%. OR, odds ratio; CI, confidence interval; PHQ-9, Patient Health Questionnaire-9.

**Table 6 nursrep-15-00279-t006:** Exercise support needs of breast cancer survivors before and during the COVID-19 pandemic.

	Before COVID-19 Total (*n* = 293)		Total (*n* = 345)		PA↓ (*n* = 226)		PA→ (*n* = 119)		*p*-Value
	*n*	(%)	*n*	(%)	*n*	(%)	*n*	(%)	
Individualized support by nurses	155	(57.0)	170	(49.3)	109	(48.2)	61	(51.3)	0.049
Individualized support by physical therapist	189	(70.3)	208	(60.3)	133	(58.8)	75	(63.0)	0.009
Individualized support by physician	143	(53.6)	177	(51.3)	115	(50.9)	62	(52.1)	0.056
Individualized support by medical team	148	(56.9)	204	(59.1)	135	(59.7)	69	(58.0)	0.036
Follow-up by telephone	135	(50.2)	141	(40.9)	94	(41.6)	47	(39.5)	0.043
Follow-up via e-mail	115	(42.8)	147	(42.6)	105	(46.5)	42	(35.3)	0.012
Providing information through brochures	205	(74.5)	244	(70.7)	168	(74.3)	76	(63.9)	0.086
Providing information via video	161	(61.0)	222	(64.3)	147	(65.0)	75	(63.0)	0.259
Providing information on our website	179	(67.5)	237	(68.7)	161	(71.2)	76	(63.9)	0.151
Providing information at lectures	189	(70.0)	182	(52.8)	121	(53.5)	61	(51.3)	0.147
Exercise class	201	(73.4)	229	(66.4)	148	(65.5)	81	(68.1)	0.467
Home-based exercise	-	-	298	(86.4)	200	(88.5)	98	(82.4)	0.332

Before COVID-19: September 2017–March 2018. During COVID-19: February–July 2022. Exercise support needs were measured on a 7-point scale from 1 (“Not necessary at all”) to 7 (“Very necessary”). (Percentage of responses scored between 5 and 7). A χ^2^ test was performed.

## Data Availability

The data can be made available upon reasonable request.

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
