# Peer review of "The Relationship Between Changes in Physical Activity and Physical and Mental Health in Female Breast Cancer Survivors Undergoing Long-Term Activity Restrictions in Japan"

_nursrep, 2025, doi:10.3390/nursrep15080279_

Round 1
Reviewer 1 Report
Comments and Suggestions for Authors
Comments
Thank you for your contribution. The study addresses an important topic with clear public health implications; however, several aspects of the methodology, presentation, and interpretation require clarification and improvement to strengthen the manuscript and ensure transparency and reproducibility.
Section 2.1. Study Design:
This section would benefit from a more detailed description. An important question is how the survey was applied and what methods were used to ensure the reliability of the results, especially considering that the study was conducted during the fifth and sixth COVID-19 waves. If the survey was administered in person, how were the survivors identified? If it was conducted virtually, what strategies were employed to ensure the quality and authenticity of the responses?
Section 2.2. Participants and Recruitment Strategy:
The process for conducting interviews should be explained in greater detail. Regarding inclusion criteria, were additional factors considered such as current medication, weight, or accessibility to the survey?
Body Weight Changes Assessment:
The statement "Changes in body weight before the pandemic and at the time of the survey were evaluated on a three-point scale: increased, no change, and decreased" needs clarification. How was this change monitored or confirmed, particularly if many responses were collected via email or virtual means?
Survey Structure and Duration:
An important question to address is how many items the questionnaire included, what sections it comprised, and the average time it took participants to complete the entire survey.
Statistical Analysis:
Why were both Pearson and Spearman tests used? Would it not be more consistent to rely solely on non-parametric statistics for comparative purposes, especially given the nature of the data?
Table 1 – Sex Variable:
It may be unnecessary to include the "sex" variable in Table 1 since male participants were excluded from the outset. It would have been more insightful to explore gender identity instead.
Table Structure Recommendation:
Table 1 could be divided into two separate tables: Table 1 for general demographic characteristics, and Table 2 for data related to physical activity.
Variable Clarification – “ActivityWeight Change Before and During the COVID-19 Pandemic”:
The meaning and purpose of this variable should be better explained. Given that all participants may have had different dietary habits, what is the intended interpretation of this variable?
Formatting – Use of “N” or “n”:
The use of "N" or "n" should be consistent throughout the manuscript. Preferably, use lowercase n in italics.
Table 2 – Statistical Tests:
Please specify which variables were analyzed using Chi-square tests and which were analyzed using the Student’s t-test.
Sedentary Time Variable (min/week):
For the variable "Sedentary time (min/wk)", the reported p-value is 0.009, but the means and standard deviations appear to be quite broad. Are you confident that the statistical test used was appropriate for these data?
MET Variables (Vigorous, Moderate, Walking):
For these variables, it would be advisable to check for the presence of outliers that might be causing the large observed differences. If outliers are present, consider justifying or adjusting for them.
Table 3 – Same Formatting Comments:
Apply the same formatting comments regarding n and p-values in Table 3.
Table 3 – Scale Clarification:
For the item "I get nervous or anxious when I see news or talk about COVID-19 on social media (1–5)", it is unclear how a mean ± SD and n (%) were derived. If this item reflects the percentage of participants selecting a certain score, would a cross-tabulation approach not be more appropriate?
Table 4 – Odds Ratio or Risk?:
For Table 4, it would be helpful to clarify whether odds ratios (ORs) or risk ratios are more appropriate, and justify the chosen method accordingly.
Established Therapies – Clarification Needed:
The concept of "individualized support" needs to be more clearly defined. What do the authors mean by established therapies in this context?
Table 5 – Time Frame Distinction:
In Table 5, it should be clearly indicated which values refer to "before" and which to "during" the pandemic. Is it possible to conduct a comparative analysis between these two time periods? This type of comparison could greatly enhance the study's contribution.
Keywords – Depression Inclusion:
If "depression" was included as a keyword, the manuscript should explain why. How was depression measured, and which scales or tools were used?
Contextual Comparison with Determinants in Mexico:
The article would be significantly enriched by incorporating a comparative perspective with studies that examine similar determinants in a Mexican context. For instance, integrating findings or discussing parallels with the study published under DOI 10.3390/curroncol31110543 could offer valuable insights and enhance the relevance of the discussion for local public health implications.
Reviewer 2 Report
Comments and Suggestions for Authors
Dear authors,
First of all, I would like to congratulate the authors for “Relationship between changes in physical activity and physical and mental health in breast cancer survivors during the COVID-19 pandemic restrictions in Japan”. Therefore, in order to improve your paper I have some comments for you.
- Please, in the abstract, consider introducing a brief description of the background before writing the objective.
- The content of this section is well presented, provides relevant information, and helps the reader understand the context. Just as a recommendation, I think it would be interesting to define what constitutes regular exercise. Frequency or intensity could be defined to establish general recommendations regarding physical exercise.
- Notice a typo error in page 3, line 92. Notice another typo error in the next line (“width”).
- Exclusion criteria should not be the opposite of inclusion criteria. They should be deleted or reworded.
- It is necessary to specify at the end of section 2.2 the basic ethical issues such as informed consent, the ethics committee, etc.
- Figure 1 should be restructured to a more academic organization and presentation and applying the modifications related to the exclusion criteria.
- Regarding demographic data, how were the items selected? What sources did the authors use to define the variables?
- Then, were the changes in PA based on IPAQ levels? This needs to be clarified.
- According to the way in which the instruments are written, it seems that some of the study areas arise from ad hoc questionnaires (AP, sociodemographic data, demand for exercise support...). This can lead to confusion. I recommend that the authors provide further justification for the selection and design of these instruments.
- Notice a typo error in page 9, line 256.
- The limitations section is very brief and has forgotten important issues such as the limitations of the methodology used (causality cannot be established in a cross-sectional study, for example), the fact that the pandemic period itself has affected the participants for other reasons and has had an impact on their mental health (loss of autonomy, loss of family members, confinement...), and other issues that should be reviewed by the authors.
After reading your manuscript, I think this is a very interesting topic, and of great interest due to it contributes to create an evidence base that encourages health professionals to consider exercise counseling in this population, but it could surely also be extrapolated to other types of patients. In the manuscript it would also be interesting if you highlighted the importance of this data obtained in the form of a recommendation for future similar situations. I hope my recommendations help you to improve this manuscript. I wish you the best.
Reviewer 3 Report
Comments and Suggestions for Authors
The topic of the study is very important regarding breast cancer survivors. The purpose of the study was given as:
This study evaluated the relationship between changes in physical activity (PA) 12 and mental and physical health in breast cancer survivors during the COVID 19 pandemic restriction in Japan.
However, the researchers evaluated COVID fear, QOL which included symptoms, physical role, etc and a number of other factors The purpose of the study does not reflect all of the variables that were analyzed and discussed in the results. The results indicated differences between groups with a t test, Pearsons, etc but the results are not clearly presented by variable. If there are multiple purposes to the study, these need to be clearly stated rather than combined into general terms. For example, QOL is not mentioned in the purpose.
The other factor which is rather irrelevant in 2025 is COVID 19. It is rather late to be publishing data about factors of the pandemic. Preparing a study with clear purposes and eliminating reference to the COVID pandemic would be much more timely.
Round 2
Reviewer 1 Report
Comments and Suggestions for Authors
The authors have done an excellent job, and the overall quality of the manuscript is commendable. However, there are still some aspects that need to be addressed. Specifically, Table 2, although it contains valuable information, is quite extensive. I recommend dividing it into smaller tables to better fit the journal’s formatting guidelines. Additionally, incorporating the following references would further strengthen the discussion section:
-
https://doi.org/10.3390/curroncol31110543
-
10.1016/j.aprim.2025.103253
-
10.1097/DCR.0000000000001489
-
10.1016/j.gaceta.2013.09.002
Author Response
Thank you very much for reviewing this manuscript. We also appreciate your insightful comments. We learned a lot from them. Thank you.
We divided Table 1 into smaller tables and cited the paper https://doi.org/10.3390/curroncol3111054 as a limitations of the study, [ Although it has been reported that sexual dysfunction affects mental problems and QOL in female breast cancer survivors [37], this study did not examine the relationship with sexual dysfunction. –page12, study limitations, line6].
Reviewer 2 Report
Comments and Suggestions for Authors
Dear authors,
You have done a very good job. I would like to congratulate you.
Author Response
Thank you very much for reviewing this manuscript. We also appreciate your insightful comments. We learned a lot from them. Thank you.